# C-3PO: Towards Rotation Equivariant Feature Detection and Description

**Abstract.** Despite the recent advances in local feature matching, dealing with affine distortions remains a major challenge. While state-of-the-art methods have shown to perform well in the absence of rotation perturbations, some computer vision applications, such as object tracking and image stitching, require keypoint extraction methods that maintain high performance regardless of the image orientation. Current approaches perform extensive data augmentation to artificially acquire a degree of rotation equivariance. However, this does not only induce redundancy in the learned feature representations, but also does not provide any geometric guarantees. To address this issue, this work explores an alternative approach that instead instills rotation equivariance inside the model itself. Leveraging recent advances in group equivariant deep learning, we propose C-3PO, a family of feature detection-and-description models based on steerable group convolutions. We evaluate our method against prior work, and find that it outperforms its non-equivariant counterparts for most rotation perturbations. However, presumably due to the task's inherent sensitivity to interpolation artifacts, extending a discrete rotation equivariant model to a continuous variant provides only marginal performance gains.

**Keywords:** Feature detection and description, local feature matching, rotation equivariance, steerable CNNs.

## 1 Introduction

Image correspondence, *i.e.* determining which parts of one image correspond to which parts of another image, is a fundamental problem in computer vision[1]. Typically, correspondence is framed as detecting and describing similar points of interest (*keypoints*) in the given pair of images. This forms a foundation of several computer vision applications such as 3D reconstruction [11], structure-from-motion [24], and visual localization [23].

Classical methods, such as Scale-Invariant Feature Transform (SIFT) [14], address this problem by incorporating handcrafted heuristics within the model to identify robust keypoints. While these methods have shown to work well, they have been increasingly outperformed by recent deep learning based approaches

---

[1] As once said by Takeo Kanade when asked about the three most fundamental problems of computer vision: 'Correspondence, correspondence, and correspondence!'.

that instead *learn* to recognise suitable keypoints [9, 19, 22]. However, these approaches perform extensive data augmentation to address affine distortions such as rotations, which does not provide any geometric guarantees such as rotation equivariance.

In line with the recent developments in the field of geometric deep learning (as popularised by [4]), we aim to improve local feature matching approaches by introducing geometric priors to a deep feature detection-and-description model. Specifically, we introduce rotation equivariance to the model using steerable group CNNs, where we consider a discrete rotation group $C_n$ and continuous rotation group SO(2) using steerable CNNs. In summary, we make the following contributions:

- We propose C-3PO (Correspondence, Correspondence, Correspondence between Points in different Orientations), a family of rotation equivariant deep feature detection-and-description models, and show the implications of using group convolutions on architectural design and computational costs.
- We perform a comprehensive empirical analysis to highlight the viability of group convolutions for image correspondence. We evaluate these results through both a quantitative and qualitative analysis to study the effect of using equivariant layers compared to a non-equivariant baseline.
- We investigate the impact of both discrete and continuous group convolutions on local feature matching tasks to study the matching performance across several flavours of rotation equivariance.

## 2  Background

### 2.1  Theory

**Group theory** A group $(G, \cdot)$ is a pair of a set $G$ and binary operation $\cdot : G \times G \to G$ under which the set is closed, that satisfies the group axioms (associativity, identity, and inverse, see Appendix A.1). Groups are used to describe the symmetries of objects, such as the changes we can apply to an image without changing its semantics. A set $X$ is called a $G$-space if it is equipped with a group action based on $G$ (Appendix A.1). Two important groups in this research are the cyclic group $C_n$ of all rotations of $\frac{1}{n} \cdot 360°$, and the 2-dimensional special orthogonal group SO(2) of all planar rotations.

For two $G$-spaces $X, Y$, a function $f : X \to Y$ is called *equivariant* to $G$ if applying a symmetry transformation $g \in G$ and then computing the function $f$ produces the same result as computing the function $f$ and then applying the transformation $g$. Similarly, such a function is called *invariant* to $G$ if transforming an element does not change its function value. Formally, the two can be written as

$$\text{Equivariance: } f(g.x) = g.(f(x)), \tag{1}$$

$$\text{Invariance: } f(g.x) = f(x). \tag{2}$$

**Group CNNs** Standard convolutional networks are equivariant with respect to translations, but not to other transformations such as rotations [7]. Group convolutional networks (G-CNNs) [20] are a common tool to introduce such equivariances by not only determining a response for each translated pixel, but also for each rotated pixel as described by $G$:

$$[k \star_G f](g) = \int_{x \in \Omega} k(g^{-1}x)f(x)dx, \tag{3}$$

for some kernel $k$ and function $f$ defined on domain $\Omega$ (*i.e.* the image). In the case of a $32 \times 32$ image, introducing 4 rotations gives a tensor response of shape $32 \times 32 \times 4$, thereby 'lifting' the image to a higher domain. After a series of such convolutions, where each next convolution integrates over the entire previous group, it is common to perform a rotation-invariant pooling. In the case of rotations, the average over all rotation maps is taken to find a final representation of our image invariant to the initial rotation.

**Steerable group CNNs** When using G-CNNs, the number of responses is expanded by convolving over a larger domain, for example a roto-translation group, rather than just a translation group. An alternative to obtaining more responses on the image is extending the codomain, *i.e.* assigning higher dimensional responses to each input feature corresponding to different rotations. To ensure that our model is still equivariant to rotations, these feature vectors need to be transformed with the image. This property is referred to as *steerability*, and the resulting feature vector is referred to as a *fiber*. To assure that a convolution is steerable, the kernel $k$ needs to satisfy the *steerability constraint* (see Appendix A.2). Similar to the standard group convolution, the arising fiber for some pixel describes the response for each rotation in the rotation group. Using this, a steerable CNN is obtained by connecting a series of steerable group convolutions together.

**Continuous steerability** Using a feature for each rotation comes with the immediate disadvantage of only being able to consider a finite number of filters. By realising that assigning a feature value to each possible rotation implicitly defines a signal on the circle $S^1$, we can leverage circular harmonics and write that signal as an infinite series of complex exponents. Using a finite subset of our Fourier coefficients, the signal $s(x)$ can be approximated as

$$s(x) \approx \sum_{n=-N}^{N} a_n e^{inx}, \tag{4}$$

for some $N \in \mathbb{N}$. This allows to indirectly describe the responses for each input pixel for all rotations using a finite set of Fourier coefficients per point, solving the initial problem.

## 2.2   Related Work

**Deep feature matching** Feature matching based on robust interest point detection and local feature description is at the core of several computer vision algorithms such as image stitching [1], visual localisation [18], structure-from-motion [25, 28], and 3D reconstruction [30]. Classical approaches such as SIFT [14], ORB [21], and SURF [3] devise handcrafted features to achieve invariance to local geometric and photometric transformations in a two-stage manner: detection and description. Modern CNN-powered approaches are based on jointly learning keypoint locations and their descriptions. For instance, R2D2 [19] predicts reliability and repeatability maps to extract not only repeating keypoints, but also those that are reliable for downstream matching. Several recent approaches also leverage transformer-based attention models for feature matching [12, 22, 26]. However, these deep neural methods are not robust to rotations. One solution for ensuring rotation equivariance is using data augmentation, as is done in methods such as RoRD [15].

**Rotation equivariant feature matching** Unfortunately, data augmentation leads to redundancy, since the network has to learn to be robust to each augmented transform. Alternatively, we could introduce geometric priors to the model architecture itself as is done in rotation equivariant CNNs [6, 8]. While rotation equivariant feature extraction has not yet received a lot of attention within the field of deep feature matching, some papers have very recently started to pick up on this concept. Notably, [5] replaces the CNN backbone of LoFTR [26] with steerable CNNs based on discrete groups. While their work is primarily focused on extending a transformer-based architecture to be rotation equivariant, our work instead looks into the benefits of applying steerable group convolutions to a *fully-convolutional* network architecture. Closest to our work, [17] extends upon the R2D2 architecture by replacing its fully-convolutional backbone by a $C_8$ equivariant one, and mainly looks into combining their architecture with others to create an ensemble with high coverage of different rotation angles. In contrast, our work focuses on pure equivariant models, and studies the effects of instilling different levels of rotation equivariance, from the discrete case to the continuous case.

## 3   Methodology

In order to make local feature matching robust to rotations, we introduce geometric priors to the model directly. For this purpose, we propose C-3PO, a family of novel deep feature detection-and-description models based on steerable group convolutional networks. As illustrated in Figure 1, each network takes an RGB image $\mathbf{I} \in \mathbb{R}^{H \times W \times 3}$ as input, and produces (i) a set of dense $D$-dimensional feature descriptors $\mathbf{X} \in \mathbb{R}^{H \times W \times D}$, (ii) a repeatability map $\mathbf{S} \in [0, 1]^{H \times W}$, and (iii) an associated reliability map $\mathbf{R} \in [0, 1]^{H \times W}$. In the remainder of this section, we introduce these models along with the baseline models we used in our experiments.

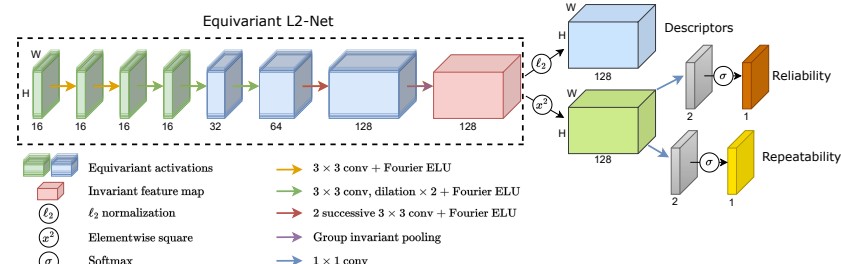

Fig. 1: **C-3PO Network Architecture.** The network architecture of the SO(2) variant of C-3PO. The initial layers comprise an equivariant variant of L2-Net. In line with [19], the remaining part of the network consists of three heads outputting the feature descriptors, repeatability map, and reliability map.

**R2D2 baseline** As our primary baseline, we consider the R2D2 detection-and-description model for our experimental evaluations [19]. Our formulation is generic for CNN-based feature detection methods. However, for evaluating rotation equivariance, we choose the R2D2 model for its high performance and well-documented code. In addition, the R2D2 model consists of a fully-convolutional network design that enables end-to-end optimisation of both feature extraction and description. This fully-convolutional structure of R2D2 allows for direct substitution with equivariant convolutional layers. R2D2 uses a modified L2-Net [27] backbone, consisting of 7 convolutional layers with monotonically increasing channel-sizes (see [19] for details).

**C-3PO** The network architecture of C-3PO, shown in Figure 1, largely follows the R2D2 model. Analogously to [5], we alter the L2-Net backbone of R2D2 by substituting the convolutional blocks with steerable ones, which makes the backbone equivariant to rotations. Each block applies an equivariant layer, batch normalisation, and an activation function that is applied either to the signal directly, or in the Fourier domain, depending on whether the group is finite.

To prevent the basis for the block expansion of the steerable filter from being empty, we replace the three successive $2 \times 2$ convolutional blocks at the end of R2D2 by two successive $3 \times 3$ equivariant convolutional blocks. After this sequence of equivariant convolutional blocks, we apply group invariant pooling by performing a max-pooling operation within each regular field to ensure that the final keypoint descriptors of the input pixels are invariant to rotations.

We distinguish between three variants of C-3PO that are each based on a different group. Each variant takes an RGB image as input, and correspondingly, the input types of the first layer are three independent scalar fields in all cases. In contrast, the intermediate signals transform according to the regular representations of their respective group. The first two variants are based on the finite group $C_n$ for $n \in \{4, 8\}$, and the last variant on the infinite group SO(2). For the discrete variants, we can use normal ReLU pointwise activation functions to ensure equivariance, since the underlying group is finite. The SO(2)

variant instead uses a Fourier ELU, which uses the inverse Fourier transform to sample the function, applies the ELU nonlinearity, and finally recovers the Fourier coefficients by performing a Fourier transform. As argued in Section 2.1, we can approximate the Fourier transform using a finite subset of Fourier coefficients. While increasing the number of coefficients provides a better approximation of the underlying signal, it also considerably increases the computational cost to train the network. We empirically found that using 4 coefficients provides a good balance between approximation precision and computational cost.

The equivariant L2-Net backbone maintains the number of layers, but reduces the number of channels per layer. The motivation for this is twofold. First, because the equivariant layers inherently capture rotated copies of the same feature, our equivariant backbone theoretically requires less channels than the base R2D2 model. Second, the addition of the group convolutional layers drastically increases the number of parameters. Whereas the base R2D2 model required roughly 0.5M parameters, the $C4$-variant required 12M and the SO(2)-variant required roughly 60M parameters.

In line with [29], we reduce the number of channels for most of the layers such that all models have a similar number of parameters. This does not only make the increased computational cost tractable, but also allows for a fair comparison. Figure 2 shows a comparison of the various models with respect to their corresponding number of trainable parameters and average inference time per input image after reducing the number of channels.

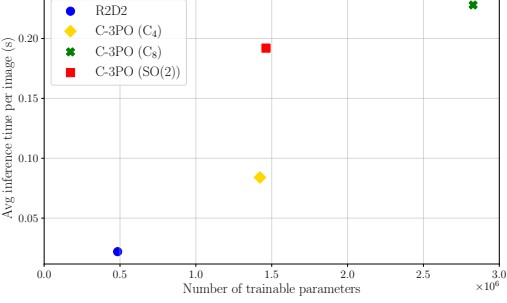

Fig. 2: **Model Efficiency.** We show the computational efficiency of different network architectures, both in terms of inference time and number of trainable parameters.

**Implementation** For our experiments, we use the original PyTorch [16] implementation of the R2D2 model[2], and modify this framework to adapt to our C-3PO model architecture. To this end, we employ the **escnn** E(n)-Equivariant Steerable CNNs library[3] to develop our rotation equivariant C-3PO variants [6, 29]. In order to enable fair comparison between our equivariant models and

[2] https://github.com/naver/r2d2
[3] https://pypi.org/project/escnn/

the R2D2 baseline, we keep the R2D2 training pipeline intact and use the same hyperparameter configuration as the original R2D2 implementation.

# 4    Experiments

**Training configuration** In accordance with the default settings for R2D2, we train for a maximum of 25 epochs using the Adam optimizer [13] with a learning rate of $1 \cdot 10^{-4}$, a weight decay of $5 \cdot 10^{-4}$, and a batch size of 8. Additionally, we adopt an early stopping mechanism where we do not continue training if the evaluation loss does not decrease for at least 3 consecutive epochs. As a result, training all the different models, including the R2D2 baseline model, required approximately 120 hours using a single Nvidia Titan RTX GPU. Performing our experiments required 16 GPU hours.

**Evaluation setup** Following [19], we evaluate local feature matching using the Homography-Patches (HPatches) dataset [2]. This dataset consists of 116 image sequences with each image sequence comprising a source image $I_S$ and $M = 5$ target images $I_{T_j}$, for $j \in \{1 \cdots M\}$. Each pair $(I_S, I_{T_j})$ is related via a homography $H_j := \mathbb{H}(I_S, I_{T_j})$, where $\mathbb{H}$ is a planar affine transformation between the two images. For a given pair of images, we first resize both to $300 \times 300$ following [17], then we use a trained model (*e.g.* R2D2) to detect and describe keypoints in each of them independently. These are then matched using a nearest neighbor search in the feature descriptor space. Next, we apply RANSAC [10] filtering to remove outlier matches. We then transform the keypoints in the first image using $\mathbb{H}$, project them onto the second image and compare with those matched in the second image to compute the metrics. A match is considered correct if its reprojection error, *i.e.*, the distance between a projected point and the corresponding point in second image, is within a certain pixel threshold. For evaluating rotation-robustness, we fix the threshold $\tau = 3$px following [19]. Our primary metric is mean matching accuracy (MMA), which computes the fraction of correct matches in a given pair, averaged across the dataset. To study the benefit of using rotation equivariant CNNs instead of standard CNNs, we compare performance in terms of MMA for input images from the HPatches dataset across rotations from $0°$ to $360°$ with an interval of $15°$.

**Quantitative results** The key results of our experiments are summarized in Figure 3. First, we show that R2D2 [19] is indeed not robust to rotations in the target image. As rotations are applied incrementally, the MMA drops steeply to zero at about $60°$. Second, we note that the $C_4$ variant significantly improves upon R2D2 at rotations in and around $\{90°, 180°, 270°\}$. This confirms the effectiveness of rotation equivariance baked into this variant at these special rotations. Third, to our surprise, both $C_8$ and steerable SO(2) variants follow a similar performance trend as $C_4$. One reason for this could be the small kernel sizes that were used, as rotating a $3 \times 3$ kernel implies only the centre pixel is

considered for the response. Among these rotation equivariant models, SO(2) does outperform $C_8$ which in turn outperforms $C_4$, but the performance gains are marginal. Further, the dip in performance across models in between the 90° rotations that are odd-multiples of 45° could be a result of rotation artifacts such as artificial edges introduced due to filling of the unoccupied regions in the rotated image.

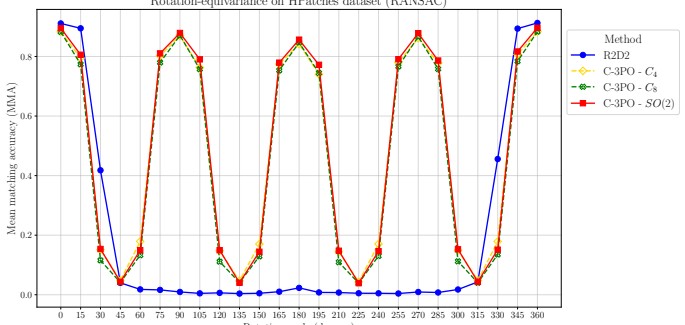

Fig. 3: **Evaluation Rotation-Equivariance.** We compare vanilla R2D2 model with our model variants that endow R2D2 with rotation-equivariance capabilities. At and around special rotations, SO(2) outperforms other variants but only marginally.

**Qualitative analysis** As a means to provide a more holistic understanding and intuition behind the quantitative results, we show feature matching results on a sample image pair from the HPatches dataset in Figure 4. All models find robust matches without any rotation of the target image, but with rotation of 90°, the vanilla R2D2 model struggles while the equivariant models are still able to find robust matches.

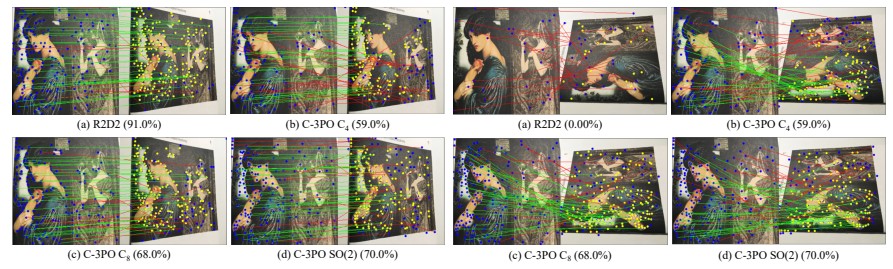

(a) Feature matching with no rotation.        (b) Feature matching with 90° rotation.

Fig. 4: **Qualitative Matching Results.** Matches found for various models for a pair of images. The percentage in parenthesis shows the fraction of correct matches for each of the models for this particular image-pair. Blue points denote keypoint detected by the model. Yellow points on the target image denote the points in source image transformed by the ground truth homography $\mathbb{H}$. Correct matches are shown in green.

Furthermore, recall that the R2D2 model jointly detects and describes salient keypoints in the image. We qualitatively analyse its ability to achieve rotation-equivariance for plain keypoint detection. To that end, we observe how detected keypoints vary when applying a 90° rotation on a sample set of images in Figure 5. Upon visual inspection, it seems keypoint detection may be equivariant to 90° rotations for all models, including vanilla R2D2. Interestingly, vanilla R2D2 seems to detect a large number of keypoints most of which are likely to be harmful for matching. This relates back to the notion of *reliability* in the R2D2 model formulation. Our model variants are not only rotation equivariant to keypoint detection but also seem to only detect *reliable* keypoints even with rotations. We observe this phenomenon in multiple samples and report more examples in Appendix B.

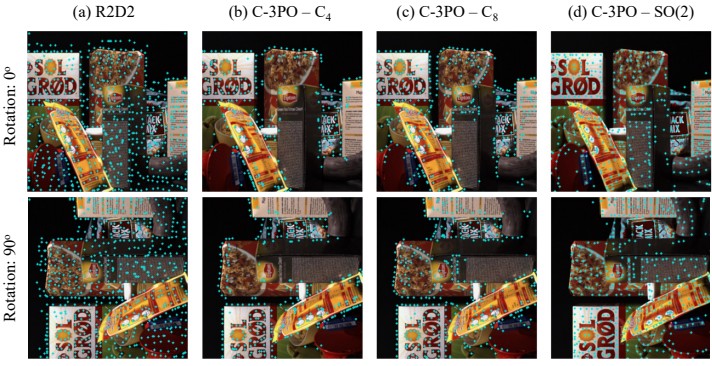

Fig. 5: **Keypoint Detection Robustness.** Qualitative evaluation of robustness of keypoint-detection against 90° rotation across all models. Our model variants detect *reliable* keypoints while being equivariant to rotations.

## 5    Conclusion

We have looked into the viability of using steerable group convolutions in feature detection and description in order to introduce rotation invariance to a high-performance feature detector and descriptor. To this end, we introduced C-3PO, a family of rotation equivariant feature detection-and-description models, and studied its robustness to rotations and homographies. Our results indicate that using the rotation equivariant models can provide additional robustness, especially for rotation angles that are multiples of 90°. However, we also found that extending the simple discrete equivariant $C_4$ model to $C_8$ and SO(2) architectures provided only marginal gains.

**Limitations & future research** While our proposed C-3PO model is able to achieve considerable performance gains when compared to our R2D2 baseline,

the experiments and model design have revealed some limitations. Firstly, converting the R2D2 model to a rotation equivariant network using steerable CNN layers came at a substantially higher computational cost than we initially anticipated, due to the large increase in required network parameters. Nonetheless, our experiments also revealed that reducing the channel sizes of the steerable layers is an effective solution to mitigate the computational cost without suffering a performance drop when compared to the baseline. Secondly, our experiments showed that, whereas using the $C_4$ variant provides a considerable performance boost for 90 degree rotations, extending the rotation equivariance to the $C_8$ and SO(2) variants provided minimal additional performance gains.

Future research should investigate why the performance difference between the discrete and continuous equivariant models is only very minimal, and how these models can actually be improved to provide reasonable robustness across all rotation angles. Some interesting directions of improvement could be (i) using larger kernel sizes for each layer to incorporate a larger context when rotating the filters and (ii) exploring different interpolation techniques when rotating.

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

# A    Additional theory

## A.1    Group theory

**Group axioms** A group $(G, \cdot)$ is a set that has an operation $\cdot$ such that two elements of the group can be combined by that operation leading to a new element, and that the following axioms hold:

- **Closure:** for all $g, h \in G, \quad g \cdot h \in G$,
- **Associativity:** for all $g, h, k \in G, \quad (g \cdot h) \cdot k = g \cdot (h \cdot k)$,
- **Identity:** there exists an $e \in G$ such that $g \cdot e = e \cdot g = g$ for all $g \in G$,
- **Inverse:** for all $g \in G$ there exists a $g^{-1} \in G$ such that $g \cdot g^{-1} = e$.

Common examples of groups are the integers under addition and the group of symmetries of a regular $n$-polygon, denoted as $(\mathbb{Z}, +)$ and $D_n$ respectively. When it is unambiguous, we refer to the group as just $G$. Moreover, we might write $gh = g \cdot h$ when it is unambiguous. If the cardinality of our group $|G|$ is finite, we call the group itself finite.

   We can equip some set $X$ with a (left) *group action* based on $G$. A group action then is a function $G \times X \to X, (g, x) \mapsto g.x$, satisfying the following axioms:

- $\forall x \in X : e.x = x$,
- $\forall a, b \in G, x \in X : a.(b.x) = (a \cdot b).x$.

One simple example of a group action is the trivial action. The trivial action simply maps $g.x \mapsto x$, which trivially satisfies the above axioms.

## A.2    Kernel constraint

The general linear group of $\mathbb{R}^n$ (notation: $\mathrm{GL}(\mathbb{R}^n)$) is the group of all automorphisms of $\mathbb{R}^n$, *i.e.* all invertible matrices $M \in \mathbb{R}^{n \times n}$. A real representation $\rho$ of some group $G$ is a function

$$\rho : G \to \mathrm{GL}(\mathbb{R}^n)$$

such that

$$\rho(g \cdot h) = \rho(g)\rho(h),$$

for all $g, h \in G$. This function associates a matrix representation to each group element in a way that the group structure is preserved. In order to assure that a group convolution is steerable, we change the way in which we convolve as follows. Instead of convolving a kernel over a larger domain (as done in G-CNNs), we convolve normally (i.e. over the group of translations) with a kernel $k : \mathbb{R}^d \to \mathbb{R}^{d_{\text{out}} \times d_{\text{in}}}$ that assigns a linear transformation to each relative position, where $d_{\text{in}}$ and $d_{\text{out}}$ denote the input and output dimensions respectively. This convolution is then steerable if the kernel can be written as

$$k(h\boldsymbol{x}) = \rho_{\text{out}}(h)k(\boldsymbol{x})\rho_{\text{in}}(h^{-1}), \tag{5}$$

for all elements $h$ of the rotation group $H$ with corresponding representations $\rho_{\text{in}}$ and $\rho_{\text{out}}$ of the input space and output space respectively.

### A.3   Matching equivariance

**Theorem 1.** *Let $M$ be a $G$-invariant matching algorithm, $F$ an equivariant feature detection, and $D$ an equivariant descriptor. Let $k_x := K(x)$ and $d_x := D(K(x))$. Then,*

$$M(d_{g.x_1}, d_{g.x_2}) = M(d_{x_1}, d_{x_2}),$$

i.e. *the matches are invariant to a transformation of the image by $G$.*

*Proof.* Let $M$ be a $G$-invariant matching algorithm, and let $K$ and $D$ be an equivariant feature detection algorithm and equivariant feature descriptor respectively. We observe that

$$
\begin{aligned}
M(d_{g.x_1}, d_{g.x_2}) &= M(D(k_{g.x_1}), D(k_{g.x_2})) && \text{(definition of descriptor)} \\
&= M(D(g.k_{x_1}), D(g.k_{x_2})) && \text{(equivariance } K) \\
&= M(g.D(k_{x_1}), g.D(k_{x_2})) && \text{(equivariance } D) \\
&= M(g.d_{x_1}, g.d_{x_1}) && \text{(definition } d_x) \\
&= M(d_{x_1}, d_{x_2}), && \text{(invariance } M)
\end{aligned}
$$

which concludes the proof.

## B   Additional Qualitative Matching Results

In this section, we present more examples of qualitative results for feature matching for our main model variants across various rotations of the target image. For reference, in Figure 6, we show results for the base case with no rotations applied on the target image. The amount of correct matches (colored in green) indicates the robustness of the model. We show the results across rotations $\{15, 45, 75, 90\}$ in Figure 7. Through these results, we also reinforce our earlier observation of equivariant models tending to detect more reliable keypoints. In this example, under all rotations, R2D2 tends to detect keypoints in the *sky* region. Such keypoints are not reliable for matching since sky is a fairly uniform region without edges and corners. All equivariant models tend to primarily detect keypoints on the buildings and less so in the sky indicating a higher degree of reliability for this points.

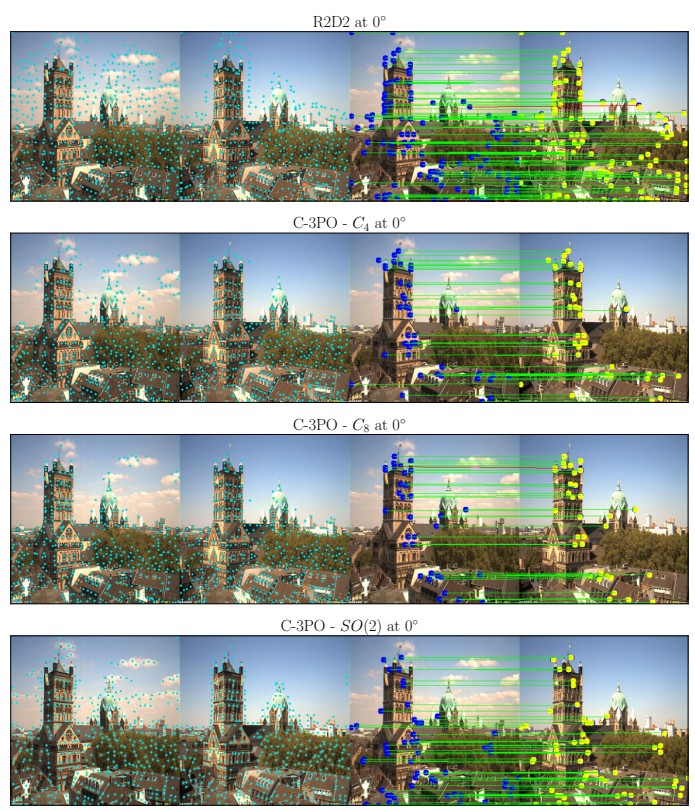

Fig. 6: **Upright Matching Results.** Base case with no rotations applied on the target image. The first two columns show the source and target images with *all* detected keypoints. The last two columns show the matches (after RANSAC) with *matched* keypoints in dark-blue, green matches denoting correct matches and yellow keypoints are obtained by projecting source keypoints using ground-truth homography.

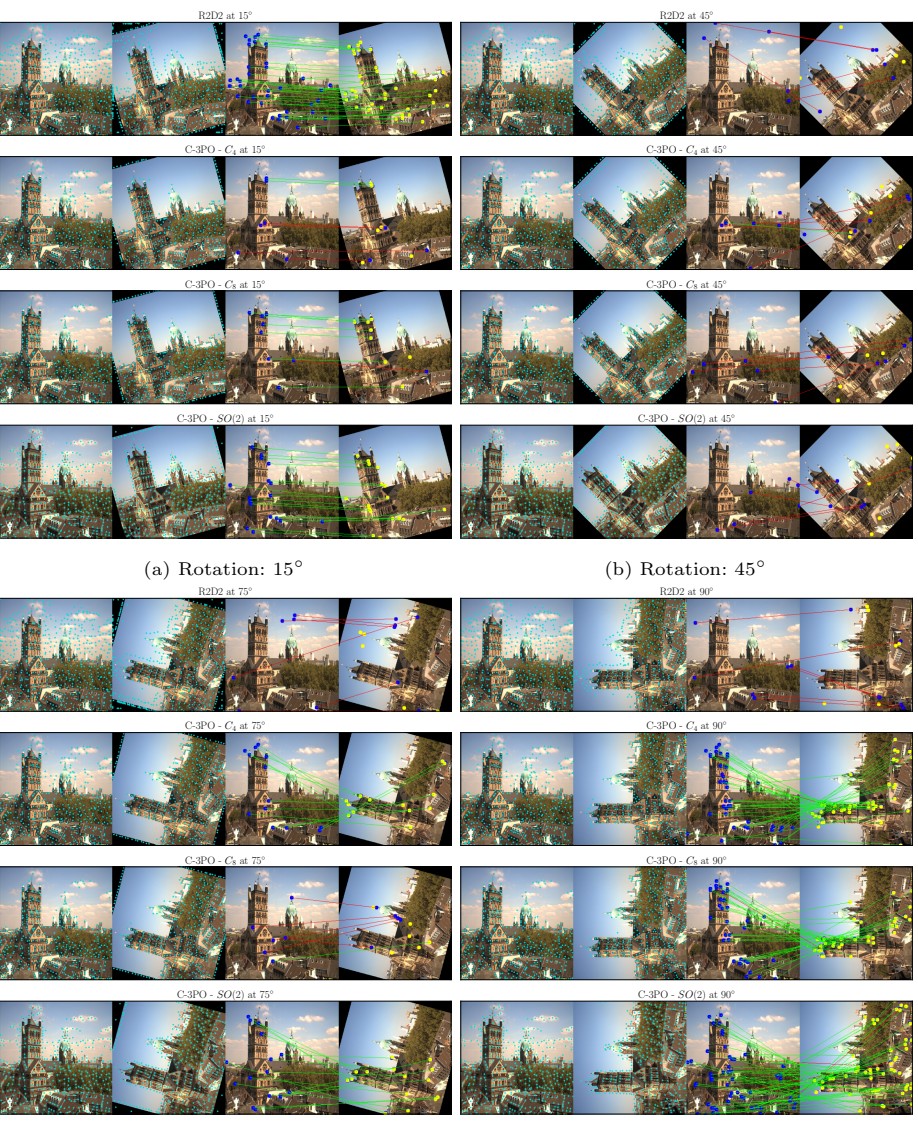

(a) Rotation: 15°          (b) Rotation: 45°

(c) Rotation: 75°          (d) Rotation: 90°

Fig. 7: **Rotated Matching Results.** Matching for rotations {15, 45, 75, 90} of the target image. The R2D2 performs comparably with the equivariant models for small rotations (15°). Both struggle at 45° rotation while the equivariant models clearly find more correct matches for 75° and 90° rotations. These results are for the sequence `i_castle` in the HPatches dataset.

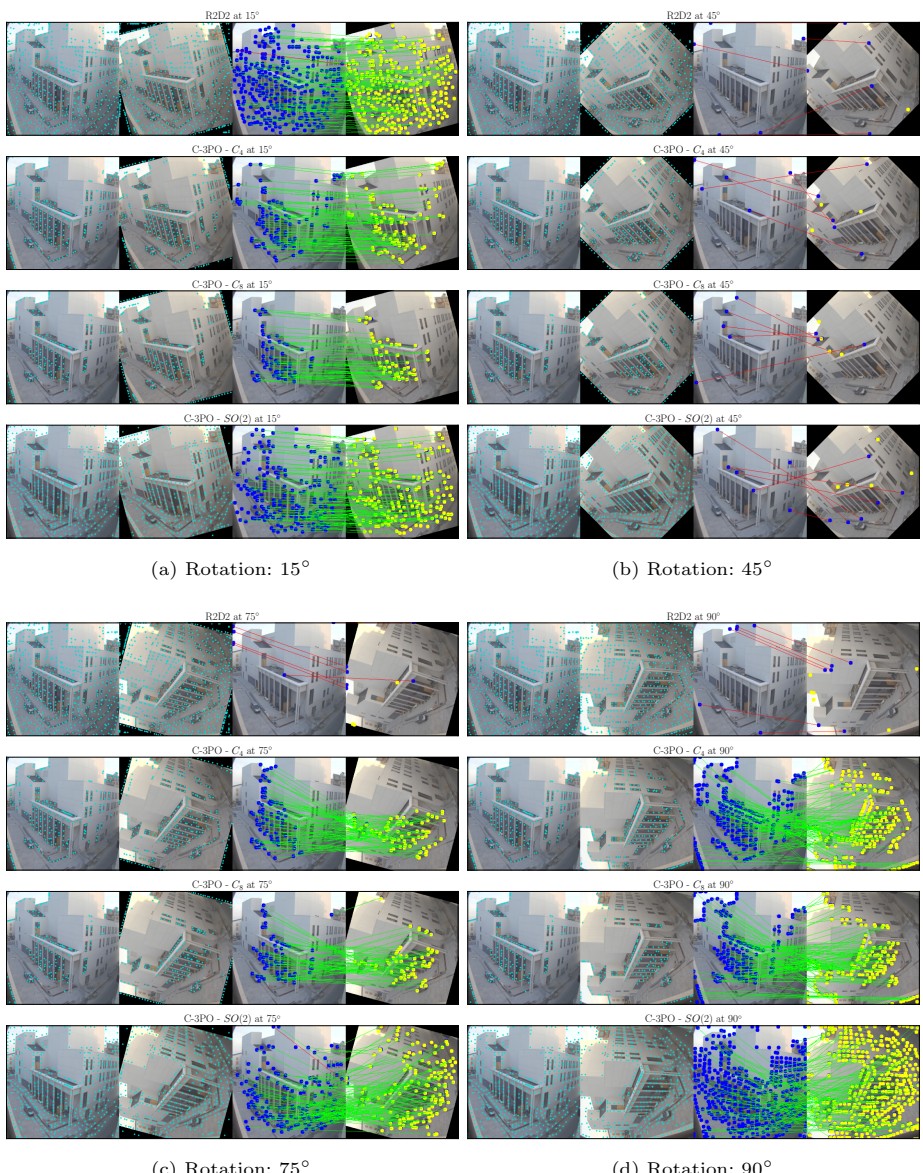

(a) Rotation: 15°          (b) Rotation: 45°

(c) Rotation: 75°          (d) Rotation: 90°

Fig. 8: **More Rotated Matching Results.** Matching for rotations {15, 45, 75, 90} of the target image. These results are for the sequence i_whitebuilding in the HPatches dataset.

## C    Rotational Stability of Different Kernel Sizes

At a rotation of $45°$, for a $3 \times 3$ kernel, 5 out of 9 cells are inliers (55.6%), whereas for a $7 \times 7$ kernel, 37 out of 49 cells are inliers (75.5%). Clearly, there is a trade-off between model efficiency and robustness to interpolation artifacts. Future research should investigate the possible interaction effects between steerable equivariant layers and the corresponding kernel sizes, and how one can pick an optimal kernel size for an equivariant CNN.

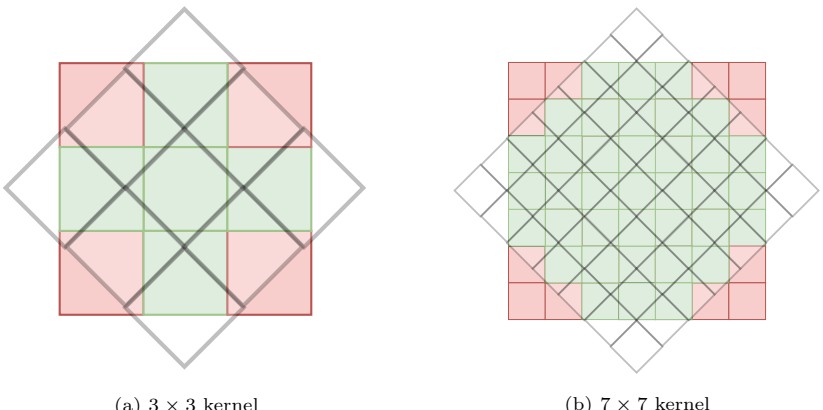

(a) $3 \times 3$ kernel                    (b) $7 \times 7$ kernel

Fig. 9: **Kernel Size Stability.** We visualise a kernel along with its rotated manifestation. Each cell of the original kernel is either marked green if this cell falls within its $45°$ rotated version, and red otherwise.

## D    Attempts to Increase the Rotation Coverage

**Using different interpolation techniques** In order to examine whether interpolation artifacts could be reduced, we experimented with different interpolation techniques in our evaluation pipeline, namely (i) nearest neighbour interpolation, and (ii) bilinear interpolation. Interestingly, the type of interpolation did not seem to affect the performance. Future research could study whether using other interpolation techniques, such as bicubic interpolation, can have a positive effect on the final performance. However, given the inconspicuous performance difference between nearest neighbour and bilinear interpolation, it is unlikely that using bicubic interpolation will have a significant effect.

**Cropping images after rotation** When rotating an image, a subset of the pixels may fall outside the image window after the rotation. This does not only mean that these keypoints cannot be used to find matches, but it may also disturb the intermediate activations and therefore the final keypoint descriptors. Therefore, when we rotate an image, we can either decide to keep the entire

original image after rotation and add black-pixel padding on the remaining corners, or we can crop a rotated section within the original image such that pixel padding is not necessary, as shown in Figure 10. Our experiments showed that using cropping before rotation increased the matching performance for all models, including the R2D2 baseline. This is likely due to the fact that cropping a section within the image reduces the number of available keypoints. Still, as the performance seemed to increase equally for all models and the relative performance differences remained roughly equivalent, it is likely that cropping before rotating is not the deciding factor for the performance 'dips' in Figure 3.

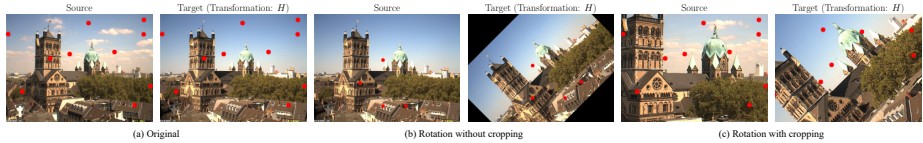

Fig. 10: **Cropping after Rotation.** Effect of applying cropping after rotation on a sample pair of images. (a) shows original sample from HPatches with dummy keypoints, (b) shows rotation without cropping, and (c) shows rotation followed by same cropping applied on both source and target images.

**Using larger rotation augmentation** We studied whether using a larger augmentation could help increase the coverage of rotation angles. Specifically, during training the input images were augmented with a random rotation of at most 20°. We hypothesised that training a rotation equivariant model with data augmentation would have a complementary effect, as the data augmentation might help around all the performance spikes of the rotation equivariant models. In contrast, the non-equivariant R2D2 model would only benefit from the rotation augmentation for small rotations. Unfortunately, our experiments showed that the rotation augmentation had a detrimental effect on the training time to convergence and the final MMA performance of C-3PO.