# OpenReview forum: "C-3PO: Towards Rotation Equivariant Feature Detection and Description"
_thecvf.com/ECCV/2022/Workshop/VIPriors — VIPriors 2022 OralPosterTBD_

### Official Review · Reviewer_x4nc · 2022-07-22
**Simple, straightforward, effective**

**Rating:** 8
**Confidence:** 4

**Review:**

The authors integrate rotation equivariant convolutions, both discrete G-Convolutions and steerable convolutions, into state of the art image correspondence networks and achieve improved generalization to unseen rotations at test time.

Strengths: simple method; excellent exposition of background, related works and method; strong empirical results; honest discussion of limitations.

Weaknesses: no real concerns. For scope, it would have been nice to see some analysis on the failures of C8 and SO(2), but a limited scope is completely fine for a VIPriors paper.

---

> ### Author Response · Authors · 2022-08-21
> **Reply to Reviewer x4nc**
>
> Thank you for the review and helpful suggestions. We are glad the reviewer acknowledges the simplicity of the method. Indeed, as the reviewer points out, we plan to extend this work into a more thorough analysis into the failure cases of C8 and SO(2) in the future.

---

### Official Review · Reviewer_rcSk · 2022-08-05
**Interesting idea, analysis missing**

**Rating:** 7
**Confidence:** 4

**Review:**

The proposed method, C-3PO, uses steerable group convolutions to achieve equivariance for the SO(2) group of continuous rotations for a learned keypoint detector based on R2D2. The method is evaluated on the HPatches dataset using rotated samples.

Strengths:
+ The method is very interesting and highly relevant in solving real-world computer vision problems. Investigating whether equivariance to continuous rotations yields any benefits compared to discrete groups in a local features setting is an excellent research question.
+ The proposed method is indeed more robust to rotations in the input. Moreover, it does seem to perform comparable to the baseline on the 0-degree setting, despite using fewer channels in the network layers.
+ The paper is clearly written and easy to read.

Weaknesses:
- It was not clear to me whether the network is still strictly rotation equivariant as it seems that the approximation of the inverse Fourier transform would prohibit this. Could the authors elaborate?
- Although the network width has been scaled down, the proposed method still contains more trainable parameters than the baseline. Why did the authors choose not to further downsize the equivariant network or add more channels to the baseline to keep the number of parameters equal?
- It is strange that the C8 and SO(2) networks have the same performance drop for rotations that are not multiples of 90 degrees. It would have been interesting to measure the equivariance error of the SO(2) network (as the MSE between feature maps of an upright and rotated input sample).
- I expected a comparison with the baseline trained on rotated input samples. It would have been interesting to see its performance since in practice that is by far the easiest solution, even though it does not provide strict equivariance guarantees. It would also have been nice to include comparisons with [5, 17].

Justification of rating: The problem setting and method are interesting and the paper is very pleasant to read. However, I would have expected some more analysis and comparisons with simple baselines.

---

> ### Author Response · Authors · 2022-08-21
> **Reply to Reviewer rcSk**
>
> Thank you for the review and helpful suggestions. We appreciate that the reviewer finds this work interesting and relevant in solving real-world vision problems. We respond to the weaknesses one by one.
> >*It was not clear to me whether the network is still strictly rotation equivariant as it seems that the approximation of the inverse Fourier transform would prohibit this. Could the authors elaborate?*
>
> - When using a approximation / sampling approach we are indeed no longer strictly equivariant. However, in practice using a relatively small number of point to sample on (which are ensured by the escnn library to be chosen as uniformly as possible over the grid) gives you a very strong signal. Our paper does not focus on this, for we already lose strict equivariance due to interpolation to the discrete grid from our continuous signal.
>
> >*Although the network width has been scaled down, the proposed method still contains more trainable parameters than the baseline. Why did the authors choose not to further downsize the equivariant network or add more channels to the baseline to keep the number of parameters equal?*
>
> - During the experiments there indeed seemed to be the trade-off between how much we downsize the equivariant network and how expressive our network is. In our implementation of C-3PO, we already severely downscaled the most parameter-heavy layers, and downscaling even more became increasingly complicated without breaking the overall structure of the original R2D2 model.
>
> >*It is strange that the C8 and SO(2) networks have the same performance drop for rotations that are not multiples of 90 degrees. It would have been interesting to measure the equivariance error of the SO(2) network (as the MSE between feature maps of an upright and rotated input sample).*
>
> - The fact that the C8 and SO(2) networks show similar performance trends is indeed surprising. We hypothesize that the inherently local nature of the task (as against a classification task that is global at the image level), makes it highly sensitive to interpolation artifacts due to rotations applied to an image. It would indeed be interesting to measure the equivariance error of the SO(2) network, and plan to extend our work into a more thorough analysis in the future, in which we compare the equivariance error across *different layers* in the network.
>
> >*I expected a comparison with the baseline trained on rotated input samples. It would have been interesting to see its performance since in practice that is by far the easiest solution, even though it does not provide strict equivariance guarantees. It would also have been nice to include comparisons with [5, 17].*
>
> - It is a fair suggestion to compare R2D2 with rotation augmentations. We tried using additional rotation augmentations with R2D2 (e.g. 15-30 degrees) and that did not seem to make a significant difference. Using more aggressive rotation augmentations degraded the performance. We do note that it was not entirely clear if some level of rotation augmentations were enabled by default in R2D2 since its data pipeline had transforms such as image-tilting.
> - Regarding comparison with [5] and [17], our aim was not necessarily to achieve a model that is competitive with state-of-the-art such as the ones mentioned by the reviewer, but rather about studying the effects of instilling different levels of rotation equivariance, from the discrete case to the continuous case. Furthermore, [5] is based on a transformer-network and it would not be a totally fair comparison with our CNN-based network. We could not compare with [17] since they do not make their code public.

---

### Decision · Program_Chairs · 2022-08-08

**Decision:**

Accept (Oral/Poster TBD)

**Comment:**

Dear authors,


Congratulations! Your work has been accepted to the VIPriors workshop. Decisions on oral/poster presentations will follow later, when the program of the workshop is finalized.

*Please note the first action item is due on Wednesday! Please see instructions below.*

**Camera-ready instructions**

There is some work left to be done to ensure your work is included in the ECCV conference workshop proceedings. The ECCV publication managers use CMT to collect all workshop papers. This means we will migrate your paper from the VIPriors OpenReview page to the centralized ECCV workshop proceedings CMT page. The VIPriors program committee will ensure the details of your work (name, title, email address) are transferred to the CMT page, after which the ECCV proceeding managers will invite you to upload the camera-ready version of your work to the centralized ECCV CMT workshop proceedings page.

Please carefully follow the following instructions:
- **Before August 10th**, ensure that the first author has a CMT account under the same email address as the OpenReview account through which the accepted work was submitted. This account will be used to invite you to upload the camera-ready paper.
- Fill out this form, to inform us that the CMT account is in order: https://docs.google.com/forms/d/e/1FAIpQLSfyAoPv2_srESKaLRHIsHoWe3Fss1Z50ykdH7SzZpenA0m_5g/viewform
- Await instructions from the ECCV publication organizers, sent through CMT, on how to submit your camera-ready paper.
- Submit the camera-ready paper **before August 22nd**. Follow the camera-ready instructions for the main conference: https://eccv2022.ecva.net/submission/call-for-papers/.

**Attending the workshop**

We invite all authors of accepted works to attend the workshop in person on October 24th 2022 at ECCV in Tel Aviv. Please note a conference registration is required to attend the workshop. The workshop will be hybrid, enabling both in-person and remote attendance. We hope all accepted works can be represented in-person by at least one author, but we understand if this is not possible. Remote attendance of the workshop will be possible, though unfortunately there are limits on presenting works remotely: we intend to enable remote oral presentations, but this is not possible for posters.

Please fill out this form *before September 26th* to inform us of your attendance: https://docs.google.com/forms/d/e/1FAIpQLSfqRhdd2pq8t4CC8hL_c8fQo_TWcbzuQH3KGLzKVE36iTW_oQ/viewform.

**Presenting your work at the workshop**

Authors of all accepted papers are invited to present a poster at the workshop. Instructions on poster format will follow at a later date, but we will ask you to print and bring your own poster to the workshop.


For more information, as well as updates on the program of the workshop, keep an eye on our website: https://vipriors.github.io.

We thank you for choosing to submit to our workshop, and we are very much looking forward to hosting you in person in Tel Aviv!


Kind regards,

Robert-Jan Bruintjes
VIPriors program committee